# The climate benefits from cement carbonation are being overestimated

Elisabeth Van Roijen[1,2], Kati Sethares[1,2], Alissa Kendall [1] & Sabbie A. Miller [1] ✉

Rapid decarbonization of the cement industry is critical to meeting climate goals. Oversimplification of direct air capture benefits from hydrated cement carbonation has skewed the ability to derive decarbonization solutions. Here, we present both global cement carbonation magnitude and its dynamic effect on cumulative radiative forcing. From 1930–2015, models suggest approximately 13.8 billion metric tons (Gt) of $CO_2$ was re-absorbed globally. However, we show that the slow rate of carbonation leads to a climate effect that is approximately 60% smaller than these apparent benefits. Further, we show that on a per kilogram (kg) basis, demolition emissions from crushing concrete at end-of-life could roughly equal the magnitude of carbon-uptake during the demolition phase. We investigate the sensitivity of common decarbonization strategies, such as utilizing supplementary cementitious materials, on the carbonation process and highlight the importance of the timing of emissions release and uptake on influencing cumulative radiative forcing. Given the urgency of determining effective pathways for decarbonizing cement, this work provides a reference for overcoming some flawed interpretations of the benefits of carbonation.

Here we model global greenhouse gas (GHG) emissions and carbon uptake from the full life cycle of cement, and we calculate the time-dependent life cycle warming potential to more accurately identify the role of GHG mitigation strategies within the cement industry. Portland cement (referred to herein as cement) is the primary hydraulic binder that holds together the constituents in concrete, one of the most-used substances in the world after water[1]. Annual cement production is projected to increase 50 % by 2050 as global population rises, along with growing needs for critical infrastructure such as housing, electricity, transportation, and sanitation[1]. Due to high levels of demand, cement production results in considerable $CO_2$ emissions driven by two key production aspects: (1) burning fossil fuels to operate high-temperature kilns, and (2) mineral-derived $CO_2$ emissions from limestone decarbonation i.e., calcination (wherein the primary raw material for cement production is converted from $CaCO_3 \rightarrow CaO + CO_2$). The emissions from cement production, coupled with the high rate of consumption, make concrete responsible for over 7 % of global anthropogenic $CO_2$ emissions and 2-3 % of global energy use[1]. Studies

have found that notable $CO_2$ emissions reductions for cement can be achieved through the use of novel materials and constituents[2–5], as well as through the use of alternative fuels for energy requirements[6,7]. However, to reach net-zero emissions for future global cement production, additional direct air capture efforts will need to be implemented[8,9]. Carbon capture utilization and storage (CCUS) is one approach that is frequently proposed for cement decarbonization, but its application is currently limited due to factors such as high costs and availability of requisite infrastructure[10].

Among the mechanisms discussed to decarbonize cement, its ability to react with atmospheric $CO_2$ has been proposed as a method to counter its production emissions. To produce materials such as concrete, cement reacts with water to form hydrate minerals that hold together aggregates (e.g., crushed rocks). Hydrated cement can naturally absorb atmospheric $CO_2$ throughout its lifecycle in a process called carbonation, where $CO_2$ diffuses into the concrete structure and reacts with hydrated cement products in the presence of pore water to form carbonate minerals. Previous studies have suggested that this

[1]Department of Civil and Environmental Engineering, 2001 Ghausi Hall, University of California, Davis 95616, USA. [2]These authors contributed equally: Elisabeth Van Roijen, Kati Sethares. ✉e-mail: sabmil@ucdavis.edu

$CO_2$ uptake potential of cement is quite substantial[11–13]. However, the long delay between the initial rapid pulse of GHG emissions to produce cement and the decadal time horizon for this uptake means that using traditional global warming potentials (GWPs), which ignore when the emissions or removals of GHGs occur, could distort our understanding of the climate benefits of carbonation.

To achieve climate change mitigation goals, it is necessary to look at processes on a global scale and incorporate the timing of carbon uptake, as well as emissions associated with demolition, to ensure that GHG reduction potentials from cement carbonation are not over-estimated. Currently, hydrated cement carbonation is being considered in the global carbon budget and in roadmaps to emissions reduction for the cement and concrete industries[14,15]. In prior assessments of the effects of this carbonation process on $CO_2$ fluxes (i.e., emissions and uptake) globally, it has been suggested that historic carbonation has resulted in nearly half of all mineral-derived $CO_2$ emissions from cement production[12]. Projecting cement demand forward to 2100 and accounting for both energy- and mineral-derived $CO_2$ emissions, it has been estimated that uptake via carbonation could result in roughly 30 % of global $CO_2$ emissions from cement production being re-absorbed[11]. However, these studies are modeled using traditional GWP accounting, which facilitates assessment as though all fluxes occur simultaneously. Simultaneous assessment is not representative for the life cycle GHG fluxes of cement, which occur over a timescale of roughly 100 years. Furthermore, these studies do not incorporate the impact of emissions associated with end-of-life processing; rather, only production-related emissions are considered. Initial counterfactual modeling efforts of individual concrete mixtures have shown that the time dependencies associated with emissions and uptake can shift our understanding of the net-impact of carbonation on the climate[16]. Saade et al.[17] performed a case study on buildings constructed and demolished from 2018 to 2050 in Quebec, Canada and showed that $CO_2$ benefits from carbonation, when using a dynamic warming potential model, would be an estimated 3–10 % of the net fluxes. However, a comparison to traditional GWP accounting methods to dynamic methods that accounts for the global built concrete infrastructure has not been conducted.

Understanding the factors that influence the rate and extent of carbonation is critical for developing an appropriate suite of strategies to decarbonize the sector. The diffusivity of $CO_2$ in concrete is dependent on various environmental factors, such as relative humidity, $CO_2$ concentration exposure, and temperature[18,19]. Each of which will vary depending on application design (e.g., indoor vs outdoor) and location. In addition, the total carbonation depth in concrete is dependent on factors such as the ratio of surface area to volume, concrete thickness, pore structure and porosity, type of cement, supplementary cementitious material (SCM) content, and duration of exposure[20]. Alterations in these parameters will drive changes in carbonation rate and magnitude. For example, increasing the surface area to volume ratio increases the rate of $CO_2$ uptake; increasing the $CO_2$ concentration of the environment can accelerate the rate of uptake over a set period of time[21]; use of SCMs can alter pore chemistry and gas permeability, typically increasing the carbonation coefficient[22]. As a result of such factors, experimental studies have shown that the carbonation rate constant for concrete can range from 0.5 – 15 millimeters (mm) per square root year (yr) (i.e., mm/yr$^{1/2}$) resulting in carbonation depths between 4.2 and 83.7 mm after a 70 year service life[13]. Therefore, for valuation of carbonation, particularly as compared to potential upfront $CO_2$ emissions mitigation strategies, it is important to have an accurate understanding and representation of such sensitivities when modeling the benefits of concrete carbonation.

Noting the role of cement composition on carbonation potential and the movements towards using more SCMs to mitigate climate damages, the effects of different SCM types and replacement levels on carbonation must be addressed. In the most common Portland cements, a mineral composition of alite, belite, tricalcium aluminate, gypsum, and tetracalcium alumino-ferrite phases are present, with the mineral portlandite resulting from hydration of this cement being the primary compound for carbonation[22]. It is well established that the inclusion of SCMs to create blended cement systems can provide improvements to long-term strength and durability compared to traditional Portland cement[23,24], and even increase the service life of concrete structures due to reduced chloride ingress[25]. However, the inclusion of SCMs also alters the pore structure and chemistry, thereby impacting the rate of carbonation. In general, it is found that increased SCM content in a cement system results in an increase in the rate of carbonation and a decreased concentration of portlandite[22]. SCMs that contribute to pozzolanic reactions offer a reduction in portlandite paired with increased calcium-silicate-hydrate minerals that commonly support a densification of the microstructure within concrete[26]. Experimental studies have shown that replacing ≥ 25 % of cement with coal fly ash or blast furnace slag or ≥ 10% of cement with silica fume, may decrease carbonation resistance; yet increasing curing temperature and time can increase the carbonation resistance of blended cements[22]. In addition to compositional shifts, there are microstructural changes that could occur during carbonation. Upon carbonation of traditional Portland cement, pore blocking by calcium carbonate, which has a larger volume than portlandite, typically results in a decrease in porosity, potentially slowing further carbonation. Although a coarsening of pore structure is common upon carbonation of blended cements, the total porosity may increase or decrease depending on the type of SCM and level of replacement[22]. We show the resulting life cycle $CO_2$ emissions for cement with up to 50% replacement with SCMs to highlight a range in carbonation effects; however, we note that use of certain SCMs and certain degrees of carbonation can lead to durability issues, and application-specific performance demands must be considered in the selection of appropriate GHG emissions reduction measures. Further, we note our models do not reflect a change in portlandite availability in the hydrated paste resulting from pozzolanic reactions.

In this work, we use the time adjusted warming potential (TAWP)[27], an alternative to the widely used GWP characterization factors[28], to calculate the effect of timing in the emissions and removals of $CO_2$ related to limestone calcination and cement carbonation throughout the cement life cycle. Much like a net present value calculation in financial accounting, the result of using a TAWP is the equivalent amount of $CO_2$ emitted or sequestered *today*, in terms of the net effects of these fluxes on cumulative radiative forcing, the same scientific basis used by the Intergovernmental Panel on Climate Change (IPCC) to calculate GWPs[28]. For more than two decades, researchers have pointed out the importance of emissions and sequestration timing, and have proposed a number of methods for better representing the effects of timing in carbon accounting methods and in global warming indicators[27,29–34]. Brandao et al. provide the most up-to-date review of these methods, and show that a number of previous methods are mathematically identical or similar in their results[35].

In this work, historic and future flows of cement, along with their anticipated category of end-use applications (e.g., buildings, civil infrastructure) and environmental conditions were modeled to calculate total carbon uptake potential on a regional (United States) and on a global scale, incorporating variables for relative humidity, $CO_2$ concentration exposures, concrete thickness, compressive strength, and SCM content to inform carbonation modeling. The potential $CO_2$ uptake of high-surface area crushed concrete at the end-of-life (EoL) and energy-related emissions associated with demolition are included to gain a more accurate understanding of the full life cycle emissions of cement and the relative contribution of $CO_2$ uptake via carbonation. For these crushed materials, we assume adequate exposure to the local

environmental conditions. Net $CO_2$ fluxes using traditional GWP are compared to using TAWP to understand how previous models have over-estimated the global warming benefits of cement carbonation. A sensitivity analysis is performed to model the impact of environmental factors and concrete constituent selection on carbonation (for example, although the model does not account for potential changes to global $CO_2$ concentrations over time as a result of climate change, a sensitivity analysis is performed to understand the impact of $CO_2$ concentration on the rate of carbonation in concrete). Various strategies that could drive reduced emissions for cement and concrete industries are summarized to identify the difference between traditional GWP benefits and cumulative radiative forcing benefits informed through the use of TAWPs.

## Results

### Cement life cycle emissions

There are various sources of $CO_2$ fluxes throughout the life cycle of cement. Figure 1a highlights the relative contributions of each life cycle stage to the magnitude of $CO_2$ emissions for 1 kg of cement used. As noted, during cement production, $CO_2$ emissions are released from mineral- and energy-derived sources, contributing ~54% and 46% to net emissions at production, respectively. For a typical building in an urban environment with a 64 year lifespan, we find that ~0.05 kg $CO_2$/ kg cement, or roughly 12% of calcination emissions, are re-absorbed during use. At EoL, assuming concrete is crushed to a particle size between 1 and 40 mm and is left exposed for roughly 3.5 months (which is the global average exposure time)[12], we find that an additional 0.12 kg $CO_2$/kg cement or 30% of calcination emissions can be re-absorbed. However, we note that the energy required for crushing waste concrete down to this size results in GHG emissions of roughly 0.1 kg $CO_2$/kg cement, nearly negating the benefits of carbon-uptake occurring during this time. When applying the environmental impact factor utilized herein for crushed concrete to the projected quantity of EoL concrete reported in Cao et al.[11]., we find that the demolition emissions amount to roughly 30–40% of the carbon uptake totals reported in that study. Given that roughly 91% of concrete at EoL is

buried, either in a landfill or in a secondary use application (such as in road bases)[12], we also examine the carbon uptake that occurs in buried, crushed concrete and observe an additional uptake of up to 0.18 kg $CO_2$/kg cement or 48% of calcination emissions within 25 years. We note that this work focuses on the primary concrete product's life cycle, which includes its EoL, but if there are additional emissions from processes used in the secondary life, they are not reflected in the scope of this analysis.

Assuming the energy generation and cement production-related emissions occur in a pulse within a year, the dynamic effects of these loadings on cumulative radiative forcing would be negligible, but the period over which carbonation occurs is longer (Fig. 1b). During the use-phase, $CO_2$ uptake occurs slowly over time, with specific rates depending on various environmental factors such as relative humidity, the presence of coatings, and $CO_2$ concentration exposure. During demolition, concrete is crushed into finer particles with high surface area, thereby significantly increasing the potential rate of $CO_2$ uptake. We examined the role of these environmental factors on carbonation rates (see Supplementary Note 1). For example, we find that concrete with an outdoor coating in a seaside environment (low $CO_2$ concentration), can result in as little as 0.15 kg $CO_2$ uptake / kg cement used throughout a 100 year lifecycle with a carbonation rate of 0.35 mm/$yr^{1/2}$ during service life. Yet under conditions with high $CO_2$ concentrations and appropriate relative humidity (e.g., indoors), concrete can absorb ~0.34 kg $CO_2$/kg cement over a 100-year lifecycle with a carbonation rate of 12.8 mm/$yr^{1/2}$. The rate of carbonation can be further impacted by concrete mixture design and member design. Use of SCMs and thickness were noted as drivers before, but we also see how shifts in porosity, that are commonly tied to different degrees of capillary voids present in high strength concrete (>35 megapascals (MPa)) and lower strength concrete (<15 MPa), affect the rate of carbonation. In these modeling efforts, due to decreased porosity in high strength concrete, there is an expected slower rate of carbonation, ~80% less $CO_2$ uptake during useful life, and 28% less over the entire 100-year life cycle (use, EoL and secondary phase) in high strength concrete compared to low strength concrete (see Source Data 1, Sheet 22).

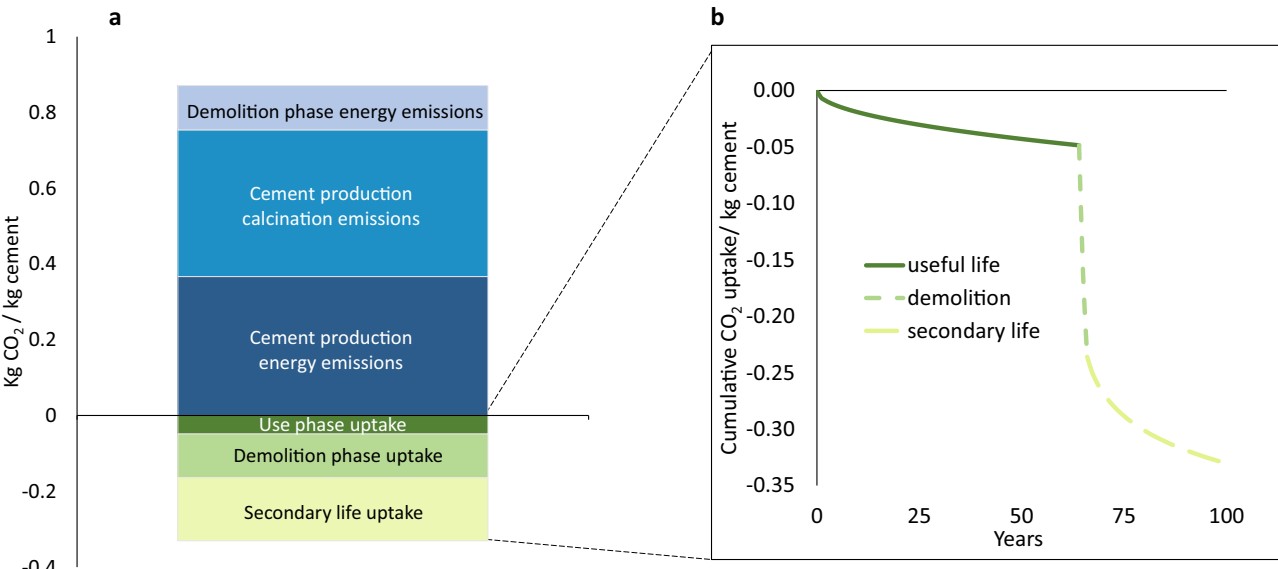

**Fig. 1 | $CO_2$ emissions and uptake per kg of cement based on global averages. a** $CO_2$ emissions per kg of cement for each life cycle stage, assuming 150 mm thick concrete with a global weighted average of strength classes. **b** Cumulative $CO_2$ uptake per kg of cement during each life cycle stage over a time horizon of 100 years. The calcination and energy emissions are based on global average cement production data. Use phase is assumed to be 64 years, demolition phase is

0.4 years, and secondary life is 35 years. At end-of-life, demolished particle size is assumed to be in the range of 1–40 mm. The concrete is assumed to be in an urban environment during useful life, industrial environment during demolition, and buried during secondary life (each affecting $CO_2$ concentration) and uncoated and exposed (driving relative humidity exposure) throughout its lifecycle. Source data are provided in Source Data 1, Sheets 12 and 13.

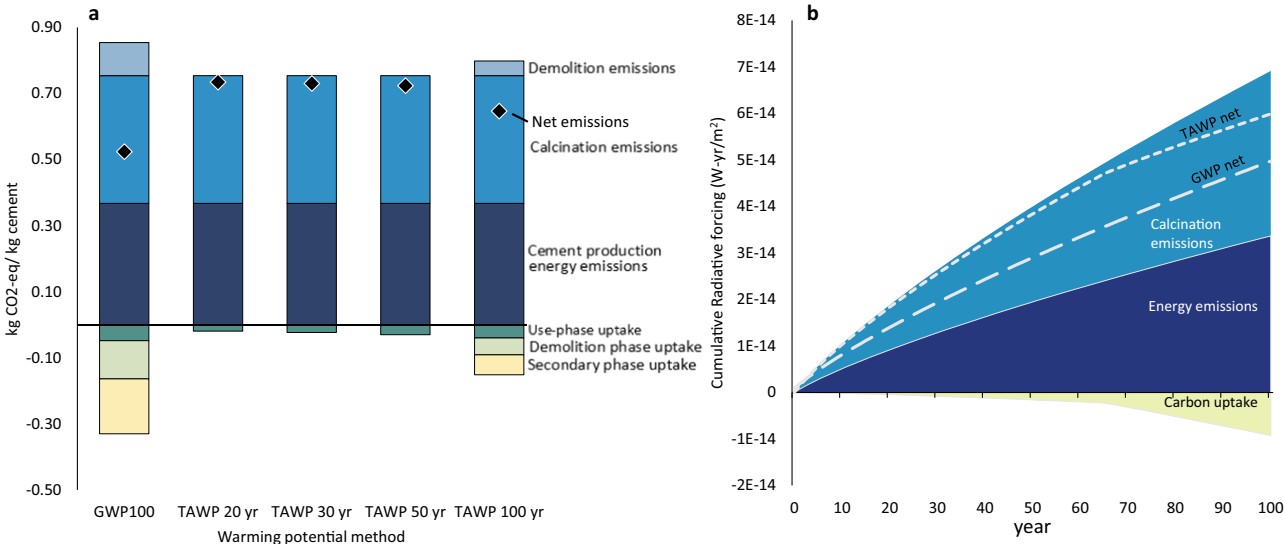

**Fig. 2 | Time dependent effects of lifecycle emissions of concrete. a** Traditional global warming potential calculated over a 100-year time horizon ('GWP 100'), compared to time-adjusted warming potentials over 20, 30, 50 and 100 year time horizons (ex: 'TAWP 20 yr'). Results are broken down by process contribution as well as total net emissions (black diamond). **b** Cumulative radiative forcing (CRF) over a 100-year time horizon. Net CRF based on traditional global warming potential approach ('GWP net') compared to time-adjusted warming potential approach ('TAWP net'). For both figures, the calcination and energy emissions are based on global average cement production data. Use phase is assumed to be 64 years, demolition phase is 0.4 years, and secondary life is 35 years. At end-of-life, demolished particle size is assumed to be in the range of 1–40 mm. For this figure, the concrete is assumed to be in an urban environment during useful life, industrial environment during demolition, and buried during secondary life (each affecting $CO_2$ concentration exposure) and uncoated and exposed (driving relative humidity exposure) throughout its lifecycle. Source data are provided in Source Data 1, Sheets 14 and 15.

## Time-dependencies

The $CO_2$ emissions associated with the production of cement, followed by the long timescale over which $CO_2$ uptake occurs is essential to understanding the climate benefits of carbonation. Figure 2 highlights the difference in results attributable to the production, use, and disposal of 1 kg of cement using traditional GWP and TAWP methods. Radiative forcing, measured in watts per meter squared ($W/m^2$), quantifies the change in energy balance of the Earth's atmosphere due to natural or anthropogenic activities, such as emitting GHGs, and cumulative radiative forcing reflects the sum of impacts from these emissions over time. Examining the difference in cumulative radiative forcing for the two methodologies (Fig. 2b), results indicate that traditional GWP methods lead to an overestimation of the benefits of carbonation by over 100%. If the impact of emissions timing is not addressed, the ability of certain climate change mitigation strategies to minimize global warming may be overestimated. Currently, concrete decarbonization goals are focused on reaching net zero emissions by 2050. While carbonation of concrete has been included in some road-maps to zero emissions[15], limited focus has been put on the impacts of the radiative forcing of emissions that will occur from now to 2050. While introducing new policies for the management of demolished concrete could enable greater carbonation at EoL, it is important for these strategies to be coupled with an assessment of emissions from demolition and techniques that reduce the production impacts of concrete in order to minimize future cumulative radiative forcing.

## Global cement-related $CO_2$ fluxes

To understand the role of carbonation on climate effects at a global scale, we pair our above findings and modeling efforts with a global material flow analysis of cement to assess the influence of past and future $CO_2$ fluxes on cumulative radiative forcing (see Fig. 3). The length of use-phase for cement-based products varied based on the application class (e.g., buildings), while the demolition phase and secondary use phase were assumed to be 0.4 years and 35 years, respectively, for all concrete (which are the global average exposure times for these life cycle stages)[12]. We project cement production emissions to 2050 based on the assumption that the GHG intensity of energy-related emissions will decrease at a rate similar to the projected decline in coal consumption (roughly 1.4 % per year)[36]. The global $CO_2$ uptake for cement used in concrete and mortar were both modeled, with relative demand for cement being 74% to concrete and 26 % to mortar[37]. We find that cumulative $CO_2$ uptake occurring from 1950 to 2050 amounted to ~46 Gt or 28 % of total emissions associated with cement production during that time frame. Note that a 100-year time horizon is used, despite having data from 1930 to 2050, in order to draw comparisons to traditional GWP, which is typically calculated using a 100-year time horizon. However, when the timing of fluxes are integrated into these comparisons, the $CO_2$ uptake during this time period results in a global warming benefit that is 67 % smaller. Furthermore, emissions from demolition over this period (assuming a demolished concrete particle size of 1–40 mm) amounts to roughly 2.5 Gt or 86 % of $CO_2$ uptake from demolished concrete during that time.

A case study was performed on the United States (US) using the same approach to exemplify the potential for scaling this method down to a region with better data granularity and/or interest in decarbonization methods. The overall $CO_2$ uptake from carbonation in the US from 1930 to 2015 amounted to 0.8 million metric tons (Mt) or 17 % of cumulative cement production emissions. Once again, considering the timing of emissions, the global warming benefit of this carbonation is 53 % smaller. It is important to note that in addition to carbonation benefits being over-estimated, the impacts of emissions production are similarly over-estimated using traditional GWP (i.e., a simple summation of emissions over this time horizon would differ from the actual effects of these emissions being produced throughout the 85 year period). Based on these projections, it is essential that cement decarbonization strategies account for this reduction in climate benefits as a result of delayed $CO_2$ uptake and not address these fluxes using conventional GWP (i.e., assuming their influence on climate impacts are occurring at an equivalent time to the emissions from

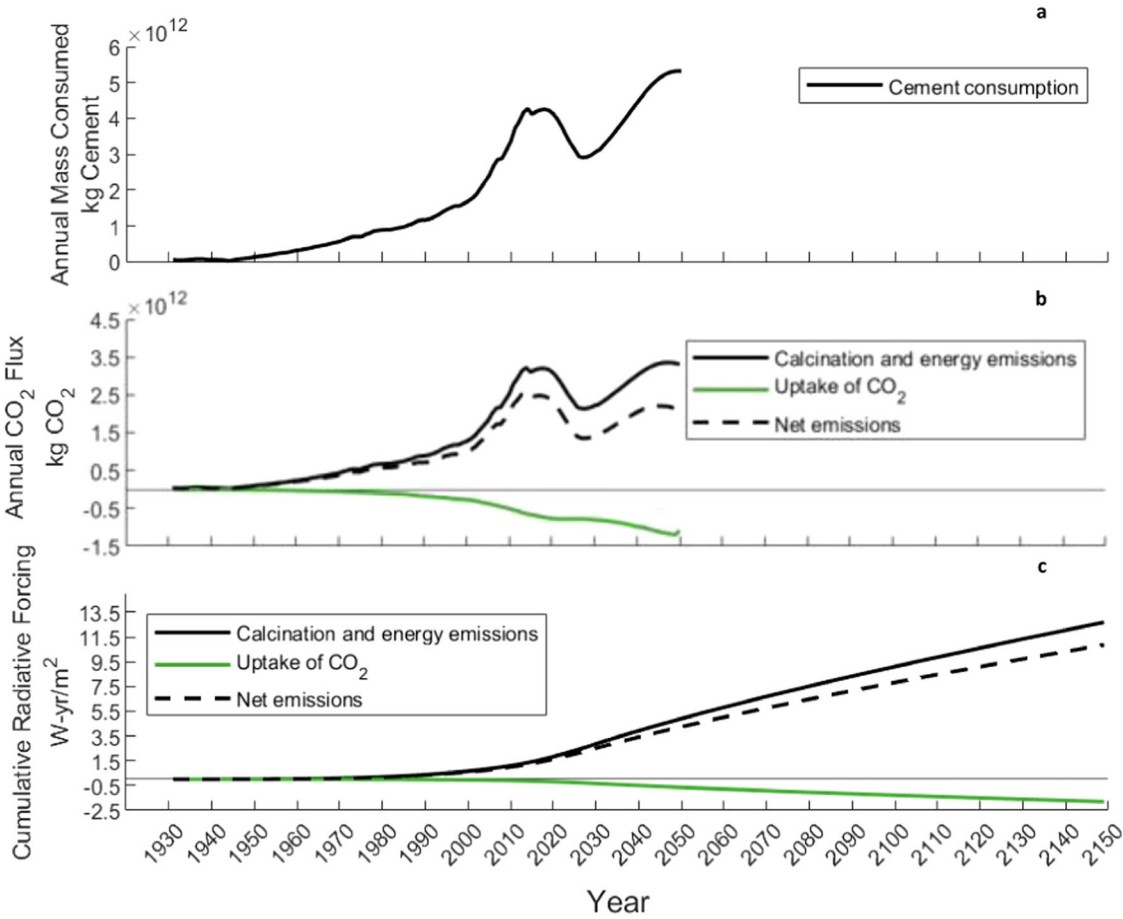

**Fig. 3 | Global cement consumption and the associated impact of emissions over time. a** Annual consumption of cement by mass in kg from 1930 – 2050. **b** Annual $CO_2$ flux for global cement consumption from 1930 – 2050 including calcination and energy emissions (black), and carbon uptake by cement (green). **c** Cumulative radiative forcing (CRF) from global cement consumption from 1930 – 2050, including CRF from calcination and energy emissions (black) and CRF from carbon uptake (green). CRF is projected until 2150 to account for the 100-year lifetime of emissions produced in 2050. Source data are provided in Source Data 1, Sheet 16.

cement production). Results also highlight the need for net-zero pathways aimed at leveraging this $CO_2$ uptake to be implemented appropriately, considering emissions from crushing and exposure conditions, to best utilize the currently in-stock concrete being taken out of use as a direct air capture mechanism and to avoid catastrophic impacts from prolonged cumulative radiative forcing from cement emissions.

**Sensitivity to mixture changes and end-of-life management**
Noting the prevalence of proposed increased SCM use to reduce emissions from cement systems production and proposed reduced particle size of demolished concrete to support carbonation, we examine the sensitivity of these factors on cumulative radiative forcing from production, use, and disposal of 1 kg of cement. The use of TAWPs is especially critical for decarbonization strategies that maximize $CO_2$ uptake at end-of-life, where we find that the apparent $CO_2$ savings are typically twice as high as the actual climate benefit over a 100-year time horizon when considering effects on cumulative radiative forcing: increasing the length of the demolition phase from 1 day to 3 months results in a 10 % reduction in GWP but only a 5 % reduction in TAWP, and exposure for 1 year can reduce GWP and TAWP by 19 % and 9 %, respectively (See Source Data 1, Sheet 20). While benefits are less pronounced than often assumed with GWP calculations, it is still important that EoL carbonation efforts are maximized for existing infrastructure as this strategy is a low-cost method and easily implementable[38]. However, it is important to note the trade-offs

examined herein for crushed concrete carbonation. Although crushing concrete to smaller particle sizes increases the surface area thereby supporting a more accelerated rate of carbonation, it also requires more energy for crushing[39], which can lead to additional energy-derived GHG emissions. Therefore, we see that crushing down concrete to a smaller particle size such as 1–10 mm, only starts to create a desired $CO_2$ uptake flux at a demolition phase length of roughly 6 months, whereas crushing down to a particle size of 1–5 mm does not provide benefits even after 1 year of exposure based on current emissions estimates (Fig. 4a).

It is also important to consider the timing of emissions when considering the role of SCMs on cement decarbonization. If selected and proportioned properly, SCMs could both accelerate the carbonation process and reduce cement production-related emissions. Although the inclusion of SCMs may result in higher rates of carbonation during use phase and end-of-life, the majority of the benefit on cumulative radiative forcing comes from the upfront savings in production-related emissions, contributing to 34–78 % of total TAWP reductions (Fig. 4b). We find that introducing either pozzolanic or cementitious SCMs at a replacement rate of 25 % or more can result in the biggest TAWP reduction of 24–53 %, compared to end-of-life mitigation strategies (See Source Data 1, Sheet 20). Given the need to transition to net-zero emissions by 2050, incorporating policies that drive reduced material-production emissions, such as increased SCM use, is imperative. Increased utilization of SCMs should also be feasible given that all of the largest cement producing countries are currently

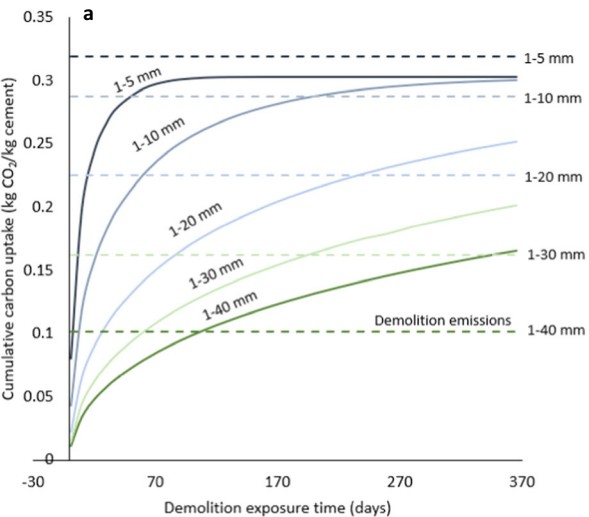

| SCM content | % Reduction in production emissions (TAWP100) | % Reduction from use phase (TAWP100) | % Reduction from EoL (TAWP100) | Total % reduction |
|---|---|---|---|---|
| 7.5% limestone | 7.5 | 5 | 6 | 19 |
| 7.5% silica fume | 7.5 | 7 | 8 | 22 |
| 15% limestone | 15 | 6 | 7 | 28 |
| 15% silica fume | 15 | 7 | 7 | 29 |
| 25% slag | 25 | 5 | 6 | 36 |
| 25 % fly ash | 25 | 9 | 7 | 41 |
| 50% slag | 50 | 5 | 5 | 59 |
| 50% fly ash | 50 | 9 | 5 | 64 |
| No SCMs | 0 | 5 | 7 | 12 |

Lowest GHG reduction — Highest GHG reduction

**Fig. 4 | Sensitivity of cement lifecycle emissions to demolition and production processes. a** Cumulative carbon uptake occurring at end-of-life of concrete as a result of demolition. For this figure, it is assumed that the concrete is exposed and uncoated. Various particle diameters are considered. Demolition emissions (dotted lines) associated with crushing the concrete down to the respective particle diameters are also included. **b** Summary table of greenhouse gas reductions relative to cradle-to-gate Portland cement production (reported in terms of time-adjusted warming potential) occurring during production, use, and end-of-life stages as a result of varying supplementary cementitious material (SCM) replacement levels. Note that the rate of carbonation for each SCM is based on the best available data at the time of modeling. These data are based on a meta-analysis of experimental studies[22], which does not account for different water-to-cement ratios, cement compositions, and gradations, all of which can impact how SCMs alter micro-structure and their ability to contribute to densification[64]. Further, this figure does not reflect a change in portlandite availability in the hydrated paste resulting from pozzolanic reactions. Source data are provided in Source Data 1, Sheets 17 and 18.

capable of generating SCMs at amounts greater than half of their total cement production[40]. However, it is important to note that the global supply of SCMs, such as coal fly ash and granulated blast furnace slag, may decrease in the future if net-zero emissions pathways are followed. This expectation is because these are industrial byproducts from high-emitting industries. For coal fly ash, pathways are suggesting the phase-out of coal plants by 2040[41], and for blast furnace slag, pathways are suggesting a shift from traditional blast-furnace to electric-arc furnace steel production[42,43]. If these decarbonization pathways are implemented, the change in supply availability may result in the need for alternative waste materials to be utilized as SCMs[44] to avoid the need for increased use of natural minerals or increased transportation distances[45]. A range of SCM replacement levels of up to 50 % was examined. However, it is important to note that at replacement levels of 50 %, some SCMs such as fly ash or slag could result in a reduction in material performance[46]. Performance characteristics that could be altered through use of these SCMs must be examined prior to implementation, and only appropriate levels and compositions of SCMs should be applied to maintain necessary functionality of built systems.

## Discussion

Future work should examine the sensitivity of varying global concrete recycling rates, as well as secondary use applications, on overall CO₂ uptake. Here, assumptions for the secondary life of concrete remain constant, assuming a 35-year use phase in a buried environment (such as use in road-based applications) for all concrete. However, not all concrete will undergo a secondary use phase; some countries such as the US recycle up to 75 % of demolished concrete while other countries such as South Africa landfill roughly 90 % of concrete waste[47]. In addition, studies have shown that waste aggregate from concrete can be substituted for natural aggregates in concrete at a rate of 20–30 % without inhibiting performance, thus highlighting the potential for waste concrete to be used in higher-strength applications[48]. Utilizing recycled concrete will likely be an important aspect of decarbonizing the cement and concrete industries.

It is important for future work to consider the concomitant human health impacts associated with concrete demolition processes. In addition to energy-related GHG emissions, the process of crushing concrete will result in dust that may contain dangerous levels of hazardous substances such as crystalline silica, nickel, cobalt, lime, gypsum, and chromium compounds[49]. Studies have found that the elemental composition of dust from demolished buildings consists of 45 % lime, and -10 % silica, which can result in serious lung diseases such as silicosis[50,51]. In addition, the machinery required to crush and transport the concrete will also result in local particular matter pollutants[52].

The uptake of CO₂ by cement over the useful life of the cement product (e.g., concrete, mortar) has been modeled by several authors; however, our work shows the significance of integrating the dynamic nature of these emissions and uptake to determine the magnitude of climate benefits from this CO₂ uptake mechanism. Without integrating the time-adjusted effect of emissions on the atmosphere, the potential role of this mineral carbonation on contributing to decarbonization goals can be grossly overestimated. This global-scale analysis provides perspective on the magnitude of cumulative radiative forcing benefits that could be achieved through carbonation of concrete. Factors integrated into this numerical assessment included: (a) various environmental factors, such as relative humidity, CO₂ concentration exposure, and temperature; (b) various designs of mixtures and component factors, such as surface area to volume ratio, member thickness, pore structure and porosity (in these efforts, these are linked to compressive strength), and SCM content; as well as (c) user and EoL management factors, such as duration of exposure, crushed concrete particle size, and secondary life. Further, commonly discussed emissions-reduction strategies of crushing concrete at EoL to benefit from the effects of a greater surface area on concrete carbonation and improving concrete mixture designs through shifts in SCM use were examined. We find that for many end-of-life strategies where carbon uptake is accelerated, the time-dependent climate benefits are roughly half as much as the CO₂ savings on a per-kg of cement scale. We find that the presence of SCMs has a larger impact on overall TAWP due to the reduction in production emissions resulting in long-term

cumulative radiative forcing benefits, coupled with the potential for accelerated carbon uptake. This work creates a unique and critical foundation to assess how mitigation strategies can be tailored to drive desired carbon uptake wherever possible in both new and existing infrastructure and to ensure that the beneficial effects of $CO_2$ uptake in decarbonization efforts are not overestimated.

## Methods

### Goal and scope of assessment

The goal of this study was to examine the carbonation effects, including time-adjusted warming potentials, to understand effects on cumulative radiative forcing and mechanisms to drive desired carbonation to mitigate climate burdens. The environmental impact analysis was conducted on a global scale, examining the cradle-to-grave (production, use and end-of-life) carbonation effects of concrete. Various concrete components were analyzed and incorporated to encapsulate the carbonation effects including cement content, SCMs, coarse aggregates, hydraulic lime and chemical admixtures. The impacts of carbon uptake were examined on a granular scale (per kg of cement) and on different regional scales (US and global).

Here, we use equations for calculating carbon uptake in concrete that we modified based on equations developed by Xi and adapted by Cao[11,12]. Modifications were made to equations so the dynamic timing factors of uptake can be considered in the three life cycle phases where carbonation occurs: useful life, demolition phase, and secondary use phase. Additionally, rather than utilizing an average value or distribution for the various parameters that determine the carbonation rate of concrete, we model changes occurring over time as a result of shifts in the cement market (e.g., the percent of cement used in certain applications, which in turn impacts parameters that drive carbonation, such as the thickness and service life, and changes in the carbon-intensity of energy). Further, we leveraged data from a recent meta-analysis of carbonation[22] to integrate the effects of various SCMs on the rate and magnitude of carbonation for concrete mixtures. The total carbon uptake of cement was calculated on a basis of 1 kg of produced, used, and disposed cement to allow for a more detailed analysis of the total uptake occurring at each life-cycle stage and the impact of different factors on the rate of uptake. The impact of different parameter variations on the carbon uptake can identify which variations have the most potential to increase carbon uptake in concrete. Global and US cement production data were combined with concrete end use and longevity data to model the historic impacts of carbon uptake, project future impacts, and assess the time-adjusted effect of regional cement consumption on the atmosphere. The regional models offer perspective of the historic performance of concrete in service and the rate of $CO_2$ savings carbonation can achieve relative to different scales of cement consumption. The modeling efforts required to analyze the production statistics and examine carbonation effects are outlined in the subsequent sections.

### Cement production and consumption

Cement consumption data are used wherever available to calculate carbon uptake relative to actual cement used in each region. The exception is global demand, where it is assumed the difference between production and consumption is marginal. Historic cement production data from 1900 to 2019 for the United States have been compiled by and are available through the United States Geological Survey (USGS) National Minerals Information Center[53]. A combination of historic (1931 – 1949)[54] and projected (1950 – 2050)[11] global production data are used to estimate apparent consumption. Global cement production and consumption can be seen in Source Data 1, Sheet 3. The amount of cement being used for concrete and mortar applications was assumed to be 74 % and 26 % respectively, based off of the data presented by European Ready Mixed Concrete Organization (ERMCO) 2017[55].

### Greenhouse gas emissions from concrete

As Portland cement and blended Portland cements are the dominant cementing material used in concrete, which in turn drives global cement production, we focused on this class of cements (as opposed to more rarely used cements, such as calcium sufloaluminate or magnesium-based cements). Life-cycle cement emissions were calculated to include those from limestone decarbonation (calcination), thermal energy from kilning (with the baseline considering a pre-calciner/preheater kiln), and electricity emissions from quarrying, raw meal preparation, finish grinding, and cooling. Electricity and energy demand for US cement production were calculated using data from the Portland Cement Association and Getting the Numbers Right (GNR) 2016 data, respectively[56]. Emissions associated with US thermal energy demands for cement production were calculated using the US 2015 fuel mix[57]. The US electricity mix was averaged from all state mixes. Calculated emissions can be seen in Source Data 1, Sheet 1. For global scale modeling, electricity and thermal energy demand for cement production was obtained from GNR 2016[58]. The thermal energy fuel mix, and electricity mix for global average cement production were obtained from GNR 2016[58] and the International Energy Agency (IEA) 2016[59], respectively. For both global and US regional analyses, starting in the year 2023, energy-derived emissions per kg of cement were assumed to decrease at a rate of 1.4 % per year to 2050, in line with the predicted 40% decrease in coal consumption by 2050[36]. US and global electricity and energy assumptions for years prior to 2023 can be found in Source Data 1, Sheet 1 and 2, respectively. Calcination emissions per kg of cement were calculated using stoichiometry; namely, cement was modeled with a clinker containing a ratio of 65 % lime originating from limestone, and cement was modeled assuming a 80 % clinker content with 5 % gypsum and 15 % additional interground mineral additives (note: for the US, 95 % clinker and 5 % gypsum were modeled, as mineral additives are more commonly used at the concrete batching stage than the cement production stage for this country). Total US and global cement emissions per year can be seen in Source Data 1, Sheet 1 and 3, respectively.

To model emissions tied to the demolition process and particle size of crushed concrete, several additional modeling assumptions were made. GHG emissions related to concrete demolition are dependent on various factors, such as the type of fuel used for machinery operation, efficiency of crusher, transportation distances, and desired crushed particle size diameter. Therefore, an average of 15 GHG values obtained from the literature is used to capture the potential variation in global emissions associated with demolition (see Source Data 1, Sheet 21). Furthermore, it has been found that energy-use associated with concrete demolition increases with decreasing particle size: Nedeljkovic et al.[39] report a 3-fold increase in energy consumption for crushing concrete to 5 mm diameter particles compared to 25 mm. Given that energy-requirements are responsible for the majority of demolition emissions[60], this relationship was therefore utilized to estimate emissions for crushing concrete to diameters of 1–40 mm, 1–30 mm, 1-20 mm, 1–10 mm, and 1–5 mm (see Source Data 1, Sheet 17).

### Cement content

A single average cement content is used to model all concrete mixtures. This number may be varied by end use market or over time, but insufficient data were available to provide varying cement contents by end use market. For average cement content data see Source Data 1, Sheet 5. For mortar applications, an average cement content of 284 kg/m³, obtained from Xi et al.[12] was utilized. Average cement content in the US was calculated as 277 kg/m³ using the ERMCO cement content statistics from 2001–2018[61]. The calculated global average cement content is 302 kg/m³, found by averaging ERMCO cement content statistics from 2001–2018 for 21 countries[61]. The average cement contents from each country were weighted by the

percentage contribution of each country to total cement production using historic production data from the USGS, 2001–2018[53].

## Phase durations

The useful life of cement varies based on its end use. Eight end use categories with distinct service lives and percentage share of the US market were identified for cement used in the United States; see Source Data 1, Sheet 7[62]. The market share was used to divide yearly cement consumption by end use and apply a service life to each end use category. The mean service lives vary between 45 and 90 years, but the largest percentage of the US market, the streets and highways sector, also has the shortest mean service life. The same approach is used to apply a useful life to global consumption data, but only three end use categories are used: residential, non-residential, and civil engineering. The global market percentages were estimated based on historic consumption data. The global average for demolition phase duration is used for all cement and is 0.4 years[12]. The secondary life for all cement is assumed to be 35 years, which is the estimated secondary useful life in the US[62]. While the world secondary life duration is greater at 60 years, a more conservative estimate is used because the model does not consider variations in how the cement is used during its secondary life.

## Factors affecting the carbonation coefficient

This modeling effort is based around utilization of Fick's law, and as such, carbonation of concrete is controlled by a carbonation coefficient. The carbonation coefficient ($K$) is a function of four individual coefficients that modify the carbonation rate based on the relative humidity ($Bec$), whether the concrete is coated ($Bcc$), the atmospheric $CO_2$ concentration based on the location of the concrete ($BCO2$), and whether supplementary cementitious materials (SCMs) have been used to replace cement in the mixture ($Bad$), as seen in Eq. 1. The carbonation of concrete is divided into three phases ('i'): the useful life, demolition, and secondary use phases. Our modeling adaptation that differentiates these phases facilitates an ability to consider variation in the location and exposure conditions for a single concrete product (e.g., in-use concrete can be modeled in an industrial or urban setting, while demolished concrete can be modeled as buried).

$$K_i = Bec_i * BCO2_i * Bcc_i * Bad_i \tag{1}$$

An exposure coefficient has been added to account for the effects of relative humidity on $CO_2$ uptake in concrete[12]. To determine the coefficient, the porosity of the cement (which is modeled here as a factor of compressive strength), along with the relative humidity of the environment have to be considered. For relative humidity (RH), 5 exposure conditions were considered: outdoor exposed (RH 85 %), outdoor sheltered (RH 85 %), indoors (RH 40 %), wet (RH 100 %), and buried[13,63]. The coefficients were presented by Pade and Guimaraes and primarily reflect conditions in Europe[13]. The exposure conditions were adapted for this study by taking a weighted average of the exposure conditions using a global average strength class distribution (using data from ERMCO 2017)[55]. When modeling US and global carbonation, the exposure coefficient is assumed to be the 'exposed' condition during the useful life and demolition phases and 'buried' during its secondary use phase. The exposure coefficients can be seen in Source Data 1, Sheet 11. Given that the majority (70 %) of mortar is used for rendering/plastering and decorating applications[12] and is made up of fine aggregate, it was assumed that the exposure conditions for mortar correspond to a low strength class (<C15).

A coating coefficient has been added to account for the effects of coatings and coverings applied over the concrete while in use. The coefficients account for indoor and outdoor concrete coatings and painted concrete and can be found in Source Data 1, Sheet 10[12]. When modeling US and global concrete carbonation, the coating coefficient

is assumed to be 'none' for all three phases because there are insufficient data surrounding the percentage of coated concrete on the market. However, the impact of coatings on concrete carbonation can be seen in the sensitivity analysis.

A location coefficient has been added to account for the effects of atmospheric $CO_2$ concentration on carbon uptake. For $CO_2$ concentration, ambient exposures based on atmospheric $CO_2$ concentrations that have been reported by Xi et al.[12] for 6 locations were considered: urban, rural, seaside, industrial, road and buried. The concentrations at these locations are weighted relative to natural conditions, 400 ppm, to calculate the adjustment factors[22]. The factors can be seen on Source Data 1, Sheet 9. When modeling global and US carbonation, the location factor is assumed to be 'urban' during the use phase, 'industrial' during the buried phase, and 'road' during the secondary use phase.

It has been shown that SCMs can have varying effects on the rate of carbonation of cement in concrete, as well as contribute to reductions in production-related GHG emissions, when used as a cement replacement or additive. Here, we use factors to address the role of SCM content on the rate of carbonation by leveraging a recent meta-analysis of carbonation data[22]. Five mineral additives (limestone, fly ash, silica fume, blast furnace slag, and natural pozzolans) with varying levels of replacement (up to 50 wt%) are considered. It is assumed that natural pozzolans will behave similarly to fly ash when used as a replacement or additive in cement. The SCM replacement factors can be found in Source Data 1, Sheet 8. The effects of several key parameters on carbonation are shown in the Supplementary Note 1, see Supplementary Table 1 and Supplementary Fig. 1.

**Calculating carbon uptake.** The depth of concrete carbonation in each life cycle stage was calculated using Fick's law of diffusion (Eq. 2, where $d_i$ is the depth of carbonation in mm during life cycle stage 'i'). The exposed surface area of the concrete was calculated by dividing the annual mass of consumed concrete ($W$, in kg) by the cement content (a fixed value, $Ci$, in kg/m³) and the concrete thickness (variable dependent on concrete application, $Tk$, in mm). A concrete thickness is specified for each end use category. The global average thicknesses were calculated from the US average thicknesses and redistributed into the three global utilization categories of residential, non-residential, and civil engineering. The concrete thicknesses can be seen in Source Data 1, Sheet 6. The thickness of cement used in mortar depends on the application. The thickness of mortar used in rendering/plastering, repairing/maintenance, and masonry as well as the amount of mortar being used in each application was obtained from Xi et al.[12]. Multiplying the carbonation depth by the exposed surface area thereby gives the volume of cement carbonated (Eq. 3).

$$d_i = K_i * \sqrt{t_i} \tag{2}$$

$$Vol = d_i * W/Ci/Tk \tag{3}$$

For cement used in mortar, three applications were included in this model: (1) rendering, plastering, or decoration; (2) masonry; and (3) repairing, or maintenance. The equations for determining the carbonation rate of mortar are similar to that of concrete. The primary difference in modeling assumptions is tied to the carbonation rate for mortar used in masonry applications, which is considered to be impacted by the presence of rendering applied to the wall.

The carbonation rate equation can be altered as a result of changes to the concrete member geometry between life cycle stages (e.g., from buildings to crushed particles). While the geometry of crushed concrete can affect carbonation due to a difference in surface area to volume ratio for various shapes, here, our efforts consider that

once concrete is demolished, the particles are assumed to carbonate as if they are spherical. This assumption holds during secondary life because the crushed concrete particles are typically reused the way they were demolished or are crushed further into similar-shaped particles. The carbonation depth in the useful life and demolition phases are calculated using the same equation as Xi, a modified version of Fick's diffusion law[12]. Carbonation depth in the secondary life is calculated as a total carbonation depth over the demolition and secondary life phases; see Eq. 4.

$$d_t = k_s \sqrt{t_s} + k_d \sqrt{t_d} \qquad (4)$$

The fraction of cement carbonated over the demolition and secondary life, $F_s$, is defined by Eq. 5. $F_s$ is a cumulative fraction.

$$F_s = \begin{cases} 100 - \dfrac{\int_a^b \frac{\pi}{6}(D-d_t)^3}{\int_a^b \frac{\pi}{6}D^3} & (a \geq D_1) \\[2ex] 100 - \dfrac{\int_{D_1}^b \frac{\pi}{6}(D-d_t)^3}{\int_a^b \frac{\pi}{6}D^3} & (a < D_1 < b) \\[2ex] 100 & (b < D_1) \end{cases} \qquad (5)$$

Where $d_t$ is the total carbonation depth over the demolition and secondary use phases and $D_1$ is the maximum diameter of particles undergoing full carbonation in the demolition and secondary use phases.

### Time-dependency for global warming potentials

The time-dependent global warming impacts of cement carbonation were calculated using a methodology developed by Kendall[27]. The TAWP for the full life cycle of cement (production, use, demolition, and secondary life) was calculated using Eq. 6. Traditional global warming potential is calculated using cumulative radiative forcing, which is the integral of radiative forcing over a specific time horizon. To capture the time-dependent effects of emissions, an additional variable 'y' is added which is the year at which the emissions occur. By subtracting 'y' from the analytical time horizon (AT), the actual time horizon of the emission is captured.

$$TAWP_{p,d,s} = m_{p,d,s} * \int_0^{AT-y} RF_{CO2} \, dt \qquad (6)$$

In Eq. 6, $RF_{CO2}$ refers to the radiative forcing of $CO_2$ and $m_{p,d,s}$ refers to the mass of emissions from production, demolition and secondary life respectively. The TAWP of $CO_2$ uptake during useful life, demolition and secondary life ($TAWP_{p,d,s}$ respectively) for 1 kg of cement was used to calculate the total life cycle TAWP. The full life cycle TAWP equation and detailed assumptions used in the TAWP calculations can be found in Supplementary Note 1. The cumulative radiative forcing and TAWP data are available in Source Data 1, Sheets 14-15.

### Reporting summary

Further information on research design is available in the Nature Portfolio Reporting Summary linked to this article.

## Data availability

The data generated and used in this study can be found in the Supplementary Information. Source data is provided in Source Data 1. Source data are provided with this paper.

## Code availability

The code that was used in this study to create the figures have been deposited in the Dryad repository and is available at: https://doi.org/10.5061/dryad.6hdr7sr7n.

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

## Acknowledgements

The authors acknowledge funding provided by Natural Resources Defense Council (A21-2618-0). Dr. Sabbie Miller would also like to acknowledge funding provided by the United States National Science Foundation (CBET-2033966). This work represents the views of the authors, not necessarily those of the funders.

## Author contributions

E.V.R. (Data curation, Formal Analysis, Writing—original draft, Writing—review and editing, Visualization), K.S. (Data curation, Formal Analysis, Writing—original draft, Visualization), A.K. (Conceptualization, Writing—review and editing, Supervision, Funding acquisition), S.A.M. (Conceptualization, Data curation, Writing—original draft, Writing—review and editing, Supervision, Funding acquisition).

## Competing interests

The authors declare no competing interests.
