## [Peer review file · Nature Communications]

REVIEWER COMMENTS

Reviewer #1 (Remarks to the Author):

The study focuses on understanding the effects of in-use carbonation of cementitious materials on the overall CO₂ emissions from cement production. The authors make a compelling case that, considering the maximum CO₂ absorption capacity of the concrete as a "negative" in the CO₂ cycle of the material is a (gross) simplification: emissions from cement production occur mainly during production, while carbonation is a slow process that will take decades. Furthermore, the major part of carbonation will occur at the end of life, according to the authors.

The authors convincingly show that time is of extreme importance here, and that it must be taken into account in whole-life cycle calculations. Furthermore, they acknowledge the fact (and consider it in their analyses) that carbonation is a slow process occurring from the concrete surface, so the whole structure must be considered, not a functional unit of concrete (e.g. 1m³), since not all of the material is readily "carbonatable". In terms of methodology, I don't feel qualified on judging the details, but it looks sound.

However, I am puzzled by the recommendations of the authors: they suggest increasing the use of SCMs in order to speed up carbonation (while also reducing the initial CO₂ emissions since less cement would need to be used). It is well known that, indeed, concretes with SCMs carbonate faster. However, carbonation of blended concretes also has more negative consequences compared to carbonation of OPC concretes: it increases porosity, decreases strength, and causes possible shrinkage. Furthermore, such rapid carbonation will lead to pH reduction and potentially fast corrosion of reinforcement in (reinforced) concrete structures. In such cases, repairs, or even premature replacement might be necessary. The environmental burden of these processes may not be negligible (although I am not sure quantitatively how much it would be). I am afraid that, while the proposed analyses improve upon the current practices by considering time explicitly, the functional unit may also need to be changed to somehow consider (reinforced) concrete. Otherwise, the conclusions may end up as they are in the current study- something that most engineers and scientists would know could be problematic.

Reviewer #2 (Remarks to the Author):

Reviewer's Comments:

The paper by Sethares et al models the global emission and carbon uptake from cement on a global scale taking the timing of carbon uptake into consideration. It's a nice and interesting paper but certain issues require some clarifications.

1) On page 4, line 2 the authors mention the use of an alternative approach to GWP and TAWP to quantify the effects of timing... However, as shown in Fig. 2a the timing has already been included in

TAWP so it is not clear what exactly is included in your alternative approach. Please explain and clarify for the readers.

2) Fengming Xi et al's paper (Nature Geoscience, Nov 2016) seems to indicate a similar approach to yours (i.e. see their Fig. 2). Please clarify how your approach and models differ from theirs.

3) Cumulative radiative forcing needs to be defined in your paper since some readers may not be familiar with it.

4) The authors switch between cement and concrete (i.e. secondary life of cement should be secondary life of concrete since it is the concrete that is being reused in road-based applications etc).

5) Does your model used for Fig. 1 include the different types of concrete and their different cement contents? For example, high strength concrete vs. conventional concrete.

Some additional comments:

Page 1, line 28: delete "current" since the current kilns use far more alternative fuels and less fossil fuels

Page 2, line 14: you may also want to mention that the carbonates plug the pores and hence CO₂ uptake depends on the thickness of the structure and may be rarely found in the center of thick concrete structures)

Page 3, line 2-4: what about CO₂ concentration adjustments

Page 3, line 15: may want to add porosity and pore structure of concrete as another parameter

Page 4, line 7: need to define IPCC (i.e. Intergovernmental Panel on Climate Change)

Page 4, line 11, show "that" a number of ...

Page 5, line 7: define EoL (i.e. End of Life); furthermore, it is the concrete that is being crushed not the cement

Page 6, line 1: which of these env. factors were included in Fig. 1b?

Page 8, line 2: you mean 1930-2050 instead of 1950-2050 since you are referring to Fig 3 which shows data from 1930 -2050

Page 8, Fig. 3: add a), b) and c) to the figures. In the Figure caption replace blue by green (carbon uptake is shown in green not blue)

Page 9, line 17: which scenarios are you referring to?

Page 9, line 19: crushing concrete to particle sizes of 1-5mm may not be feasible due to the additional energy required for grinding (the fine and coarse aggregates) to such small particle sizes which leads to an increase in CO₂ emission which does not seem to be included in your analysis.

Page 10, table 1: heading of table mentions "...corresponding reduction in global warming potential.." but the table does not list the reduction in GWP but the reduction in GHG.

Page 10, line 11: while SCMs would be a good solution, some SCMs, specifically the supply of Fly Ash is getting more limited (should be pointed out in your paper)

Page 11, line 7: replace cement by concrete

Reviewer #3 (Remarks to the Author):

The authors noted that traditional GWP accounting models ignored the delay caused by the slow rate of CO₂ uptake in cement and therefore overestimate the environmental benefits of cement carbonation. This paper considered the factors affecting the rate of carbonation in concrete and calculated cement carbonation magnitude on a regional (United States) and global scale. This research is significant in providing insight into the carbonation benefits of cement. Detailed comments are as follows:

1. The time factor in the carbonization process of cement was considered in the paper, but in the calculation of carbon emissions from the cement production process, data from EPA and IPCC were directly quoted. Considering that there was also a delay in the generation of energy such as electricity, please explain why the time factor of carbon emissions during production was not considered.
2. Please explain why it was assumed that "No additional material processing occurs between dismantling and secondary use."
3. The recycling of waste concrete after the demolition of buildings is currently the mainstream direction, in which concrete is generally crushed into recycled coarse aggregate (greater than 4.75 mm), recycled fine aggregate (0.15-4.75 mm) and recycled powder (less than 0.15 mm). The carbonation capacity of recycled aggregates with different particle sizes is different due to different specific surface areas as well as cement contents. Please provide additional information on how aggregate particle size is considered in the calculation process?
4. The authors have taken time into account in the calculation of carbonization, but the corresponding time-dependent calculation model was proposed by Kendall et al. Please describe the novelty of this study in terms of computational methodology.

Reviewer #4 (Remarks to the Author):

The sponge effect of CO₂ emission and capture by cement is discussed in a number of publications, but a comprehensive picture related to the specific kinetic constraints has not yet been obtained. The submitted manuscript tackles this subject, addressing especially the long-time range consideration and

dynamic environmental effects. The clarity in the manuscript's text is commendable, underscoring the significance of the work. While undoubtedly meaningful, there are a few remarks that warrant attention.

(1) In light of the extended timeframe up to 2100, it is essential to consider various factors when quantifying the potential CO₂ capture capacity. This includes the examination of different sources of electricity, optimization of calcination procedures, and addressing limitations in raw material supply. Notably, as we approach the net-zero deadline outlined in the Paris Agreement, it is anticipated that the proportion of coal and fossil-based energy in quantifying emissions from cement calcination will decrease.

(2) I suggest that the authors integrate cement chemistry into their calculations of the carbonation effects for different types of cement. While furnace slag and fly ash show promise as alternatives to ordinary Portland cement, it's important to note their distinct cement chemistries. TAWP remains robust in predicting potential mitigation, as well as the comprehensive database in L.J. Parrott's work. However, I recommend introducing a parameter from the cementitious system when quantifying the captured amount of CO₂ based on carbonation depth.

(3) Additionally, the applicability of SCMs incorporated cement in specific constructions (e.g., Water and waste management) should be considered. Specifically, the data provided in Tables SDS7 and SDS8. Can the replacement of Blast Furnace Slag and Fly Ash be feasibly extended up to 50 wt% in these constructions?

(4) Minor corrections are needed for Figure 3. The figure captions (a), (b), and (c) should be appropriately marked. Additionally, it's noted that the carbon uptake by cement is currently represented in green instead of blue.

(5) Revisions in the Methods section are recommended for better clarity and specific references. Currently, the methods lack precision, with crucial details mostly found in the supporting information.

Title: The climate benefits of concrete carbonation are being over-estimated.

Response to Reviewer Comments

The authors appreciate the constructive comments of the reviewers in helping to improve the manuscript, and we hope that the changes made address the relevant concerns.

(Note: line numbers in responses refer to revised (“marked-up”) manuscript with in-line changes)

Reviewer #1

Comment R1.1: “The study focuses on understanding the effects of in-use carbonation of cementitious materials on the overall CO₂ emissions from cement production. The authors make a compelling case that, considering the maximum CO₂ absorption capacity of the concrete as a "negative" in the CO₂ cycle of the material is a (gross) simplification: emissions from cement production occur mainly during production, while carbonation is a slow process that will take decades. Furthermore, the major part of carbonation will occur at the end of life, according to the authors. The authors convincingly show that time is of extreme importance here, and that it must be taken into account in whole-life cycle calculations. Furthermore, they acknowledge the fact (and consider it in their analyses) that carbonation is a slow process occurring from the concrete surface, so the whole structure must be considered, not a functional unit of concrete (e.g. 1m³), since not all of the material is readily "carbonatable". In terms of methodology, I don't feel qualified on judging the details, but it looks sound. However, I am puzzled by the recommendations of the authors: they suggest increasing the use of SCMs in order to speed up carbonation (while also reducing the initial CO₂ emissions since less cement would need to be used). It is well known that, indeed, concretes with SCMs carbonate faster. However, carbonation of blended concretes also has more negative consequences compared to carbonation of OPC concretes: it increases porosity, decreases strength, and causes possible shrinkage. Furthermore, such rapid carbonation will lead to pH reduction and potentially fast corrosion of reinforcement in (reinforced) concrete structures. In such cases, repairs, or even premature replacement might be necessary. The environmental burden of these processes may not be negligible (although I am not sure quantitatively how much it would be). I am afraid that, while the prosed analyses improve upon the current practices by considering time explicitly, the functional unit may also need to be changed to somehow consider (reinforced) concrete. Otherwise, the conclusions may end up as they are in the current study- something that most engineers and scientists would know could be problematic.

Response R1.1: The authors thank the reviewer for taking the time to provide productive and detailed feedback on the manuscript. The reviewer brings up an excellent point regarding the potential trade-offs that exist due to long-term durability concerns of reinforced concrete structures, particularly upon carbonation of blended cements compared to traditional Portland cement. The authors note first that our intent had not been for the strategies listed to be perceived as recommendations. We believe that the way the manuscript had been structured led to a lack of clarity around this point, so we have included revisions to address this point, as discussed below.

We agree that a shift in the hydration products due to pozzolan reactions can lead to differences in microstructure upon carbonation relative to a Portland-cement based concrete. However, the pronounced

benefits of using supplementary cementitious materials (SCMs) on reducing corrosion due to the densification of the microstructure are well established, particularly for regions prone to chloride ingress, such as coastal areas where the majority of future construction is currently expected to occur^{1,2}. We note that in general, it is seen that carbonation mechanisms are different for cementitious systems containing SCMs versus traditional Portland cement, in part due to blended cements typically having lower portlandite concentrations. The shift in microstructure and hydration products from use of SCMs results in carbonation in SCM-blended cements happening more rapidly than high Portland-clinker based cementing systems. It is possible that this set of mechanism can lead to coarsening of pore structures and reduced mechanical strength: yet the total porosity upon carbonation of blended cements can either increase or decrease depending on the type of SCM and level of replacement³. Further, based on a recent review from a International Union of Laboratories and Experts in Construction Materials, Systems and Structures (RILEM) technical committee examining the effects of using SCMs on concrete carbonation, there are also a handful of other factors that may impact the carbonation process in blended cements³. Such factors can include as curing times and temperature, which can significantly decrease the extent of carbonation in blended cements, and there is little understanding of the impact of secondary hydration products from SCMs on carbonation resistance. Our intent was not to address the complex balance of benefits and drawbacks to use of SCMs during reinforced concrete use as there are multiple compounding factors, and the expected rate of carbonation during use would likely not be the sole driving durability mechanism of concern. Rather, our work was meant to indicate that with increased proposed use of these materials to meet materials emissions goals (e.g.,⁴), we expect that upon end-of-life there will be an acceleration in the rate of carbonation, but a lower peak in the magnitude of carbonation due to less availability of portlandite. We have clarified this in the text and added several additional citations to give context to the contributions of this work as well as the scope boundaries of what is considered (Page 5, Lines 5-25; Page 6, Lines 1-9).

The authors do agree with the reviewer that SCM replacement should not be presented as mitigation strategy solely due to the increased rate of carbonation, due to the potential drawbacks of long-term durability. Particularly because it was not our intent for the manuscript to read as such, we have rewritten the “mitigation strategies” section of the manuscript to instead focus on the sensitivity of carbonation behavior in cement systems to various mechanisms, including mitigation strategies such as SCM replacement (Figure 4b). In addition, a discussion regarding the impact of SCMs on carbonation has been added to further clarify the nuances and limitations of this mitigation strategy (Page 5, Lines 5-25). The authors have also added statements to clearly highlight that, due to the above reasons, optimal replacement levels of SCMs in cement will depend on the desired mechanical and durability performance characteristics of the concrete, which will be highly application specific (Page 19, Lines 11-17).

1. Olsson, J. A., Miller, S. A. & Alexander, M. G. Near-term pathways for decarbonizing global concrete production. *Nat. Commun.* **14**, (2023).
2. Alexander, M. *Marine Concrete Structures: Design, Durability, and Performance*. (Elsevier, 2016). doi:<https://doi.org/10.1016/C2014-0-01042-8>.
3. von Greve-Dierfeld, S. *et al.* Understanding the carbonation of concrete with supplementary cementitious materials: a critical review by RILEM TC 281-CCC. *Mater. Struct.* **53**, 136 (2020).
4. Global Cement and Concrete Association. Concrete Future - The GCCA 2050 Cement and Concrete Industry Roadmap for Net Zero Concrete. *Glob. Cem. Concr. Assoc.* 1–48 (2021).

Reviewer #2

Comment R2.0: The paper by Sethares et al models the global emission and carbon uptake from cement on a global scale taking the timing of carbon uptake into consideration. It's a nice and interesting paper but certain issues require some clarifications.

Response R2.0: The authors would like to thank the reviewer taking the time to review this manuscript and provide constructive feedback on this paper.

Comment R2.1: On page 4, line 2 the authors mention the use of an alternative approach to GWP and TAWP to quantify the effects of timing... However, as shown in Fig. 2a the timing has already been included in TAWP so it is not clear what exactly is included in your alternative approach. Please explain and clarify for the readers

Response R2.1: The authors would like to thank the reviewer for highlighting the need for further clarity regarding the methods used in this study. This study utilizes the time-adjusted warming potential (TAWP) approach, as opposed to traditional GWP. This modeling method is a unique perspective to addressing the role of carbonation of concrete relative to more commonly applied methods that facilitates our addressing the effects of timing for the initial pulse of greenhouse gas (GHG) emissions associated with cement production relative to the decadal uptake associated with later life cycle stages. The phrasing of this sentence on the specified line has been restructured to emphasize that point. (Page 6 Lines 10-13).

Comment R2.2: Fengming Xi et al's paper (Nature Geoscience, Nov 2016) seems to indicate a similar approach to yours (i.e. see their Fig. 2). Please clarify how your approach and models differ from theirs.

Response 2.2: We appreciate the reviewer for asking for further clarification and emphasis regarding the novelty of this study. We note that Xi et al.⁵, as well as another study by Cao et al.⁶, both examine the global carbon uptake resulting from cement carbonation over a given time-period. However, neither of these studies consider the effects of timing for GHG fluxes on the cumulative radiative forcing tied to cement-based systems. As such, their work is limited in providing an understanding of the true benefit of this carbon uptake on the atmosphere. Rather, total uptake was summed together (e.g. Xi et al. found that 4.5 Gt of carbon have been sequestered since 1930). Xi et al. also do not examine the use of SCM's and the resulting impact on carbonation rates, which is considered herein. Finally, neither study incorporates emissions associated with concrete crushing/demolition process, which we find can be a significant source of emissions relative to the amount of uptake that occurs during this life cycle stage. As such, this work provides a notable advancement into how these fluxes are considered, particularly as they are now being monitored in assessments of our global carbon budget (e.g.,⁷) and in mitigation strategies for this industry⁴. Additional context further clarifying these points has been added to the introduction (Page 3 Lines 3-22).

5. Xi, F. *et al.* Substantial global carbon uptake by cement carbonation. *Nat. Geosci.* **9**, 880–883 (2016).
6. Cao, Z. *et al.* The sponge effect and carbon emission mitigation potentials of the global cement cycle. *Nat. Commun.* **11**, 3777 (2020).
7. Friedlingstein, P. *et al.* Global Carbon Budget 2022. *Earth Syst. Sci. Data* **14**, 4811–4900 (2022).

Comment R2.3: Cumulative radiative forcing needs to be defined in your paper since some readers may not be familiar with it.

Response R2.3: The authors appreciate the reviewer pointing out the need for more background on terms used within the manuscript. A definition of cumulative radiative forcing has been added to provide more clarity. (Page 11, Lines 1-3).

Comment R2.4: The authors switch between cement and concrete (i.e. secondary life of cement should be secondary life of concrete since it is the concrete that is being reused in road-based applications etc).

Response R2.4: The authors agree that utilizing the term “cement” when referring to secondary life can be confusing given that it is in the form of concrete at this stage. The authors have adjusted the terminology to remain consistent throughout the manuscript, by only using the term cement when referring to carbonation processes as hydrated cement is capable of carbonation but, with rare exceptions, aggregates in concrete are not.

Comment R2.5: Does your model used for Fig. 1 include the different types of concrete and their different cement contents? For example, high strength concrete vs. conventional concrete.

Response R2.5: The authors appreciate the reviewer for inquiring about the specificities and robustness of the modelling efforts used herein. Different types of concrete mixtures are included, based here on use of supplementary cementitious materials content and strength classifications. Fig 1 shows the results for the life cycle emissions and uptake of 1 kg of cement with a global average strength (we based this average on the distribution below; we provide additional context for how these strength distributions were determined in the methods). However, the model developed herein is capable of reflecting the estimated life cycle carbon uptake for both high strength and conventional concrete without taking a global average of strength classes. An example of this, as seen in the figure below (where low strength concrete refers to <C15, and high strength concrete is >C35), has been included in the supplemental materials (Data Sheet 22). Furthermore, a discussion of the impact of strength on resulting carbonation has been added to the text (Page 8, Lines 8-17).

Global Strength Class Distribution	
≤C15 (%)	12%
C16-C23 (%)	22%
C23-C35 (%)	53%
>C35 (%)	13%

Comment R2.6: Page 1, line 28: delete “current” since the current kilns use far more alternative fuels and less fossil fuels

Response R2.6: Interestingly, the dependence on fossil fuels in currently used cement kilns remains high in some locations. However, the authors agree with the critique of the word “current” in this context, which can lead to a misinterpretation by readers, so we have edited the line accordingly. (Page 2 Line 1).

Comment R2.7: Page 2, line 14: you may also want to mention that the carbonates plug the pores and hence CO₂ uptake depends on the thickness of the structure and may be rarely found in the center of thick concrete structures)

Response R2.7: The authors agree with the reviewers comment that pore blocking by carbonates can have an impact on porosity and therefore the depth of carbonation. An additional section was added to the paper that describes the impact of SCMs on carbonation rates, wherein the process of pore blocking in traditional Portland cement is discussed to highlight differences in porosity to address a point raised by Reviewer 1 (Page 5 Lines 22-25). In this location, we have also added additional context that concrete thickness is also among the list of factors that impact the depth of carbonation (Page 4 Line 17).

Comment R2.8: Page 3, line 2-4: what about CO₂ concentration adjustments

Response R2.8: The reviewer is correct, changes in atmospheric GHG concentrations can alter the relative impact on cumulative radiative forcing of 1 additional kg of a specific GHG in the atmosphere compared to another. However, this complexity also impacts traditional global warming calculation methods and is reflected in part by continuously updated Intergovernmental Panel on Climate Change reports. In this section of the manuscript, our goal was to describe the uniqueness of implementing time-adjusted warming potentials to address the fact that the GHG fluxes associated with production and use of concrete are happening at different timescales. While we believe that a discussion of the effects of GHG concentrations on warming potentials may confuse the reader, particularly for a journal such as Nature Communications, which has a broad audience, we had added several points of clarification. Namely, we addressed that concentrations of atmospheric CO₂ surrounding concrete systems can alter the carbonation (Page 4 Line 14; Page 7 Line 4,14-16) and we have highlighted that global carbon accounting, such as assessments of our global carbon budget (e.g.,⁷), have started to reflect the uptake potential of hydrated cement systems (Page 3 Lines 3-5).

7. Friedlingstein, P. *et al.* Global Carbon Budget 2022. *Earth Syst. Sci. Data* **14**, 4811–4900 (2022).

Comment R2.9: Page 3, line 15: may want to add porosity and pore structure of concrete as another parameter

Response R2.9: The authors appreciate the insight regarding useful parameters to consider in the model. We note that porosity is currently incorporated in the carbonation coefficient. To develop this modeling effort, strength class-specific carbonation rates are used, which are modeled as being correlated to porosity, as reported by Lagerblad⁸. It is possible for this relationship to become more complicated with increased use of blended cements as well as alter over time; however, it is used in this study for simplicity and because this approximation has been established in the literature as a strong proxy for carbonation modeling efforts⁶. In future efforts, we intend to develop a model that will allow a user to input mixture-unique porosity, but we believe this more highly adapted approach will be of more interest to an area-

specific audience. The authors have added porosity and pore structure to the list of the parameters in the manuscript however, to ensure that the reader understands that this variable was considered (Page 4 Line 17).

8. Lagerblad, B. *Carbon dioxide uptake during concrete life cycle – State of the art.* (2005).

Comment R2.10: Page 4, line 7: need to define IPCC (i.e. Intergovernmental Panel on Climate Change)

Response R2.10: We have corrected this initial omission of the acronym description (Page 6, Line 17).

Comment R2.11: Page 4, line 11, show “that” a number of ...

Response R2.11: We have corrected the noted typographical error (Page 6, Line 21).

Comment R2.12: Page 5, line 7: define EoL (i.e. End of Life); furthermore, it is the concrete that is being crushed not the cement

Response R2.12: We have corrected this initial omission of the acronym description. We also agree with the reviewer’s comment regarding the terminology used, and we have replaced cement with concrete (Page 7, Lines 5-6).

Comment R2.13: Page 6, line 1: which of these env. factors where included in Fig. 1b?

Response R2.13: We appreciate the reviewer highlighting the need for clarity regarding the inputs assumed for Figure 1b. All the environmental factors listed in this line of text are incorporated into Figure 1. The location (urban or industrial, etc.) is reflective of the CO₂ concentration. Similarly, the exposure conditions of concrete (whether the concrete is exposed, outdoors, covered/uncovered, or indoors) is reflective of the relative humidity of concrete. To ensure future readers understand parameters considered, these factors have been specified in the figure caption (Figure 1).

Comment R2.14: Page 8, line 2: you mean 1930-2050 instead of 1950-2050 since you are referring to Fig 3 which shows data from 1930 -2050

Response R2.14: We thank the reviewer for highlighting this discrepancy in the time horizons reported. It is correct that the figure includes emissions fluxes for the years 1930-2050 (with cumulative radiative forcing extending to 2150 due to the 100-year lifespan of CO₂). However, to compare the traditional global warming benefits to TAWP benefits associated with CO₂ uptake, a 100-yr time horizon was used. This 100-yr time horizon was selected as it is the standard time horizon applied for global warming potential. As a result, a 100-year time horizon of 1950 to 2050 was utilized to present those results. We recognize this may cause confusion to the reader. To address this point, we have included a description of this aspect of the modeling (Page 13 Lines 17-19).

Comment R2.15: Page 8, Fig. 3: add a), b) and c) to the figures. In the Figure caption replace blue by green (carbon uptake is shown in green not blue)

Response R2.15: We have corrected the noted error regarding the Figure 3 caption.

Comment R2.16: Page 9, line 17: which scenarios are you referring to?

Response R2.16: The authors thank the reviewer for highlighting the need for further clarification of the examined strategies. This line is highlighting a general trend that is observed. Namely, strategies that result in emissions reduction at end-of-life have a climate benefit that is approximately half the size of what has been calculated in the past using traditional warming potential methods. The lines proceeding this statement provide examples of such scenarios, such as decreasing concrete particle size or extending the length of exposure for demolition. We have restructured the phrasing of this line to improve clarity (Page 16 Lines 8-18).

Comment R2.17: Page 9, line 19: crushing concrete to particle sizes of 1-5mm may not be feasible due to the additional energy required for grinding (the fine and coarse aggregates) to such small particle sizes which leads to an increase in CO2 emission which does not seem to be included in your analysis.

Response R2.17: The authors appreciate the reviewer for bringing up this point. Initially, the impacts associated with concrete demolition were not considered in this study, as this practice of omission has been standard in much of the work in this area. However, we agree with the reviewer that, when analyzing the potential GHG benefits of reducing the particle diameter of demolished concrete, it is important to consider the corresponding trade-offs associated with GHG emissions that could be tied to increased energy consumption, such as that in the concrete crushing process. Therefore, to address this point, we utilize a recent study, which showed that the energy-requirements for crushing concrete down to 5mm size, vs a 25 mm size, resulted in 3 times the required energy consumption⁹. We coupled this relationship with a GHG emissions factor for concrete demolition (down to a particle size of 1-40mm), using an average of 15 values obtained from the literature (see supplemental data sheet 21 for list of sources). These efforts allowed us to provide an estimate for the GHG emissions from concrete demolition associated with each particle size considered herein (Page 7, Lines 5-8, Page 8, Lines 7-9; Page 16, Lines 20-23). A visual highlighting this sensitivity of trade-offs in energy-emissions versus uptake was also generated and included in the manuscript (Figure 4a).

9. Nedeljković, M. *et al.* Energy consumption of a laboratory jaw crusher during normal and high strength concrete recycling. *Miner. Eng.* **204**, (2023).

Comment R2.18: Page 10, table 1: heading of table mentions “..corresponding reduction in global warming potential..” but the table does not list the reduction in GWP but the reduction in GHG.

Response R2.18: We have corrected the noted error in the Table 1 heading.

Comment R2.19: Page 10, line 11: while SCMs would be a good solution, some SCMs, specifically the supply of Fly Ash is getting more limited (should be pointed out in your paper).

Response R2.19: The authors appreciate the reviewer for bringing up the topic of resource availability. A discussion of the future availability of supplemental cementitious materials, particularly industrial byproducts from high-emitting processes such as coal fly ash from coal-based electricity and granulated blast furnace slag from iron production, has been included in the manuscript (Page 19 Lines 3-8).

Comment R2.20: Page 11, line 7: replace cement by concrete

Response R2.20: The authors agree with the reviewers comment to replace the word cement with concrete and have adjusted the manuscript accordingly (Page 19, Lines 22-24).

Reviewer #3

Comment R3.0: The authors noted that traditional GWP accounting models ignored the delay caused by the slow rate of CO₂ uptake in cement and therefore overestimate the environmental benefits of cement carbonation. This paper considered the factors affecting the rate of carbonation in concrete and calculated cement carbonation magnitude on a regional (United States) and global scale. This research is significant in providing insight into the carbonation benefits of cement. Detailed comments are as follows:

Response R3.0: The authors thank the reviewer for taking the time to provide productive and detailed feedback on the manuscript.

Comment R3.1: The time factor in the carbonization process of cement was considered in the paper, but in the calculation of carbon emissions from the cement production process, data from EPA and IPCC were directly quoted. Considering that there was also a delay in the generation of energy such as electricity, please explain why the time factor of carbon emissions during production was not considered.

Response R3.1: The reviewer raises an interesting point regarding time horizons for energy generation and utilization within the cement/concrete production stages relative to the utilization and end-of-life stages for concrete. Although it is true that the acquisition of raw materials, along with energy consumption processes associated with concrete production, do not all occur simultaneously, these time factors are considered as occurring within a one-year period. Based on the effective granularity of time-dependent warming assessing, this brief time horizon does not result in a necessity for time-adjustments to ascertain climate impacts. Yet, the following decadal time horizon for use and disposal of concrete does require time-adjustments to relate the contributions of CO₂ uptake at these later life-cycle stages to that initial pulse of GHG emissions from energy resources and chemical conversions occurring in the production of cement and concrete. We have added a brief clarification of this in the text (Page 9 Lines 11-13).

Comment R3.2: Please explain why it was assumed that "No additional material processing occurs between dismantling and secondary use."

Response R3.2: The authors appreciate the reviewer inquiring about the assumption that no processing occurs between dismantling and secondary use. Here we model secondary use as a scenario in which demolished concrete aggregate is re-used in road-based applications. We examine this secondary use phase primarily to understand the maximum extent to which carbonation is achieved for the primary concrete product, as well as how long carbonation takes, depending on cement type, environmental conditions, etc. Therefore, the authors did not consider GHG emissions associated with these steps to form a secondary product. However, we note that it would be important to consider GHG emissions associated with such processes if one wanted to analyze the effectiveness of various end-of-life scenarios for concrete from a life cycle analysis perspective. Discussion regarding this trade-off was added to the manuscript for clarity (Page 8, Lines 16-18 Lines).

Comment R3.3: The recycling of waste concrete after the demolition of buildings is currently the mainstream direction, in which concrete is generally crushed into recycled coarse aggregate (greater than 4.75 mm), recycled fine aggregate (0.15-4.75 mm) and recycled powder (less than 0.15 mm). The carbonation capacity of recycled aggregates with different particle sizes is different due to different specific surface areas as well as cement contents. Please provide additional information on how aggregate particle size is considered in the calculation process?

Response R3.3: The authors agree with the reviewer’s note that the particle size and cement content of these components will affect total uptake potential and the rate of uptake. We appreciate the reviewer highlighting the need for further explanation of how aggregate particle size is considered in the calculations of the carbonation process. In this study, the carbonation depth for demolished concrete particles is calculated using the same methods as present by Xi et al.⁵ (shown in the equations below), in which particles are assumed to be spherically shaped to calculate exposed surface area. In this study, we examine carbonation depths for demolished concrete with particle sizes 1-5, 1-10, 1-20, 1-30, and 1-40 mm. However, our model could be utilized to examine particle sizes that are even smaller or larger than those examined herein. Additional context and equations have been added to the methods section for clarity (Page 31 Lines 3-13).

$$D_{0i} = 2d_{di} = 2k_{di} \times \sqrt{t_d}$$

$$F_{di} = \begin{cases} 100\% - \int_a^b \frac{\pi}{6} (D - D_{0i})^3 / \int_a^b \frac{\pi}{6} D^3 \times 100\% & (a \geq D_{0i}) \\ 100\% - \int_{D_0}^b \frac{\pi}{6} (D - D_{0i})^3 / \int_a^b \frac{\pi}{6} D^3 \times 100\% & (a < D_{0i} < b) \\ 100\% & (b < D_{0i}) \end{cases}$$

Where F_d is the fraction of demolished concrete that is carbonated, D_0 is the maximum diameter that undergo full carbonation, K_d is the carbonation rate coefficient of demolished concrete, T_d is the length of exposure in the demolition phase (in years), and D is the diameter of the demolished particles, with a and b representing the minimum and maximum values of the range specified (e.g. 1-40 mm). In this work, these sizes are used to reflect the size of a crushed hydrated-cementitious compound (i.e., not a fine layer of cement adhered to a coarse aggregate). Our models assume that aggregates do not contribute to the uptake potential of the concrete system. This point has also been clarified in the supplementary materials (Section S2).

Comment R3.4: The authors have taken time into account in the calculation of carbonization, but the corresponding time-dependent calculation model was proposed by Kendall et al. Please describe the novelty of this study in terms of computational methodology.

Response R3.4: The authors appreciate the reviewer highlighting the need for further clarification regarding the novelty of the computational analysis performed herein. The reviewer is correct that the time-adjusted warming potential model is taken from previous work by Kendall, a co-author in this work. And other authors have noted the time dependencies of GHG fluxes for cement and concrete can influence how GHG emissions are considered for 1m^3 of concrete¹⁰, or a specific region¹¹. However, a systematic quantification of the time dependent effects on a global scale has not been addressed. Considering the implications of how carbonation is becoming considered in roadmaps to decarbonization and the global carbon budget, this work derives a novel computational assessment of carbonation based

on established mechanisms for carbon uptake, new findings based on meta-analyses for the use of SCMs, and both regional and global materials flow analysis with time-adjusted warming potential. In doing so, it establishes a new foundation for consideration of these fluxes. This scaled global as well as US cement production/consumption data highlight the importance of considering the timing of emissions when determining mitigation strategies for the future. Furthermore, in addressing Reviewer Comment 2.17, this work now has yet another computational advancement beyond prior work by integrating GHG emissions from concrete demolition, which would occur decades after production emissions, but will affect the size distribution and uptake rate noted by this reviewer in Comment 3.3.

10. Moro, C., Francioso, V., Lopez-arias, M. & Velay-lizancos, M. The impact of CO₂ uptake rate on the environmental performance of cementitious composites : A new dynamic Global Warming Potential analysis. *J. Clean. Prod.* **375**, 134155 (2022).
11. Saade, M. R. M., Yahia, A. & Amor, B. Is crushed concrete carbonation significant enough to be considered as a carbon mitigation strategy? (2022).

Reviewer #4

Comment R4.0: The sponge effect of CO₂ emission and capture by cement is discussed in a number of publications, but a comprehensive picture related to the specific kinetic constraints has not yet been obtained. The submitted manuscript tackles this subject, addressing especially the long-time range consideration and dynamic environmental effects. The clarity in the manuscript's text is commendable, underscoring the significance of the work. While undoubtedly meaningful, there are a few remarks that warrant attention.

Response R4.0: The authors thank the reviewer for taking the time and for this supportive feedback.

Comment R4.1: In light of the extended timeframe up to 2100, it is essential to consider various factors when quantifying the potential CO₂ capture capacity. This includes the examination of different sources of electricity, optimization of calcination procedures, and addressing limitations in raw material supply. Notably, as we approach the net-zero deadline outlined in the Paris Agreement, it is anticipated that the proportion of coal and fossil-based energy in quantifying emissions from cement calcination will decrease.

Response R4.1: We agree with the reviewer that changes in the projected energy grid should be considered when looking at future cement production. However, in this work, future production is only modelled until 2050. The timeline on Figure 3 extends to 2150 to highlight the lasting effects of GHG emissions on the atmosphere via cumulative radiative forcing on that longer time horizon. Yet we do agree that even projections on this shorter time horizon should consider anticipated shifts in energy resources. To address this point, we have re-assessed the potential GHG uptake from carbonation relative to emissions on this time scale assuming that energy emissions associated with cement production will likely decrease at a rate similar to the projected decrease in coal consumption over the next 30 years (roughly 1.4% per year)¹² (Page 13, Lines 11-14). Figure 3, along with the supplemental materials, have been updated to reflect this data. Furthermore, we include a brief discussion on the future availability of resources such as SCMs, which are expected to shift with decarbonization efforts, in the text as well (Page 19 Lines 3-11).

12. International Energy Agency (IEA). World Energy Outlook. (2023).

Comment R4.2: I suggest that the authors integrate cement chemistry into their calculations of the carbonation effects for different types of cement. While furnace slag and fly ash show promise as alternatives to ordinary Portland cement, it's important to note their distinct cement chemistries. TAWP remains robust in predicting potential mitigation, as well as the comprehensive database in L.J. Parrott's work. However, I recommend introducing a parameter from the cementitious system when quantifying the captured amount of CO₂ based on carbonation depth.

Response R4.2: The authors appreciate the reviewer's insightful suggestion to incorporate cement chemistry into the carbonation calculations. The current modeling efforts are adapted to reflect a conventional Portland clinker (i.e., alite, belite, tricalcium aluminate, and tetracalcium aluminoferrite phases) to form hydration products and the effects of varying SCMs, accounting for their reflective compositions, are also considered. The data for the SCMs, such as the furnace slag and fly ash, come from a literature review published by a RILEM technical committee, von Greve-Dierfeld et al.³, wherein results from experimental studies all examining the change in carbonation rates for blended cements with 28 day curing stages were combined. We note these factors may not have been as clear as desired in the first draft of this manuscript. Therefore, to address this reviewer's comments and others, an additional section has been added to the manuscript to further describe the nuances surrounding the impact of SCMs on carbonation and resulting concrete performance characteristics (Page 5 Lines 5-25).

Comment R4.3: Additionally, the applicability of SCMs incorporated cement in specific constructions (e.g., Water and waste management) should be considered. Specifically, the data provided in Tables SDS7 and SDS8. Can the replacement of Blast Furnace Slag and Fly Ash be feasibly extended up to 50 wt% in these constructions?

Response R4.3: The authors appreciate the reviewer inquiring about the feasibility of SCM replacement in civil infrastructure applications such as water and waste management. Studies have shown that significant levels of SCM replacement of up to 50% fly ash, or up to 90% for slag, can be utilized and still achieve desired compressive strength characteristics (35 MPa)¹³. Studies have also shown the potential for SCMs such as blast furnace slag to be used up to 60% in ultra-high performing concrete¹⁴. However, the authors absolutely agree that these replacement ratios are not appropriate for all applications. Because this modeling effort was conducted at a global scale, the levels of replacement are averaged out over various applications and do not reflect the specification of any one mixture, any one structure, or any one location. However, we agree this point should be better stated in the work. We have added some clarifying statements to make clear to the reader that the optimal level of SCM replacement in a given application will depend on the desired performance characteristics (Page 19 Lines 13-17).

13. Brinkman, L. Environmental Impacts and Environmental Justice Implications of Supplementary Cementitious Materials for Use in Concrete Environmental Impacts and Environmental Justice Implications of Supplementary Cementitious Materials for Use in Concrete. **2021**.
14. Park, S.; Wu, S.; Liu, Z.; Pyo, S. The Role of Supplementary Cementitious Materials (SCMs) in Ultra High Performance Concrete (UHPC): A Review. **2021**, 1–24.

Comment R4.4: Minor corrections are needed for Figure 3. The figure captions (a), (b), and (c) should be appropriately marked. Additionally, it's noted that the carbon uptake by cement is currently represented in green instead of blue.

Response R4.4: We have corrected the noted error in the Figure 3 caption.

Comment R4.5: Revisions in the Methods section are recommended for better clarity and specific references. Currently, the methods lack precision, with crucial details mostly found in the supporting information.

Response R4.5: The authors appreciate the reviewer's comment and agree that adding more detailed information to the methods section of the manuscript would be helpful for the readers understanding. The majority of detail for the Methods and data used were originally in our supplementary information documentation. To address this reviewer point, we have moved a portion of the details from the supplemental materials to the methods and added further details where appropriate (Pages 22-32). Further, we intend to publish the code used to perform the work in a public repository (Dryad) to ensure reproducibility.

REVIEWERS' COMMENTS

Reviewer #1 (Remarks to the Author):

Based on the revised manuscript I am happy to recommend it for publication.

Reviewer #2 (Remarks to the Author):

I reviewed the revised version of the manuscript and accept the response of the authors to my review/suggestions. The manuscript is now ready to be published.

Reviewer #3 (Remarks to the Author):

The author responded well to the reviewer's comments and currently I have no more comments.

Reviewer #4 (Remarks to the Author):

I think that the authors have adequately addressed the comments made by the reviewers in the revised version of the manuscript.

Title: The climate benefits of concrete carbonation are being over-estimated.

Response to Reviewer's Comments

Reviewer #1 (Remarks to the Author): Based on the revised manuscript I am happy to recommend it for publication.

Reviewer #2 (Remarks to the Author): I reviewed the revised version of the manuscript and accept the response of the authors to my review/suggestions. The manuscript is now ready to be published.

Reviewer #3 (Remarks to the Author): The author responded well to the reviewer's comments and currently I have no more comments.

Reviewer #4 (Remarks to the Author): I think that the authors have adequately addressed the comments made by the reviewers in the revised version of the manuscript.

Response: The authors greatly appreciate the reviewers taking the time to review this manuscript, and we appreciate their thoughtful feedback that allowed us to improve our work.